# Improving Efficiency Assessment of Psychiatric Halfway Houses: A Context-Dependent Data Envelopment Analysis Approach

**DOI:** 10.3390/healthcare8030189

**Published:** 2020-06-28

**Authors:** Chien-Wen Shen, Chin-Hsing Hsu, For-Wey Lung, Pham Thi Minh Ly

**Affiliations:** 1Department of Business Administration, National Central University, Taoyuan 32001, Taiwan; cwshen@ncu.edu.tw; 2Calo Psychiatric Center, Xinpi Township, Pingtung County 92541, Taiwan; 3SocialTech Research Group, Faculty of Business Administration, Ton Duc Thang University, Ho Chi Minh City 700000, Vietnam; phamthiminhly@tdtu.edu.vn

**Keywords:** data envelopment analysis, rehabilitation, efficiency evaluation, chronic psychiatric patients

## Abstract

This study proposes the approach of context-dependent data envelopment analysis (DEA) to measure operating performance in halfway houses to enable suitable adjustments at the current economic scale. The proposed approach can be used to discriminate the performance of efficient halfway houses and provide more accurate DEA results related to the performance of all halfway houses in a region or a country. The relative attractiveness and progress were also evaluated, and individual halfway houses’ competitive advantage and potential competitors could be determined. A case study of 38 halfway houses in Taiwan was investigated by our proposed approach. Findings suggest that fifteen halfway houses belong to the medium level, which can be classified into a quadrant by examining both their attractiveness score and progress score. The results can be used to allocate community resources to improve the operational directions and develop incentives for halfway houses with attractive and progressive values, which can reduce the institutionalization and waste of medical resources caused by the long-term hospitalization of patients with mental illnesses. Our proposed approach can also provide references for operators and policy makers to improve the management, accreditation, and resource allocation of institutions.

## 1. Introduction

Half a century has passed since psychiatric treatments were institutionalized in the 1960s. Large sanitariums and centralized care have always been viewed as a violation of the current trend of caring models [1]. Community psychiatric care can be divided into two models: the hospital and community-based medical service models. Such an approach aims at systematic, integral, and continuous treatment and rehabilitation. Therefore, the establishment of halfway houses provides continuous and complete medical care for patients with chronic mental disorders, strengthens the community rehabilitation of patients, and improves the utilization rate of beds, thereby reducing the institutionalization and waste of medical resources caused by the prolonged hospitalization of patients with mental disorders. The main objective of the halfway house is to provide various methods for temporary accommodation and guardian services to homeless patients with mental illnesses, thus enabling their smooth return to community life [2]. Study results have indicated that patients living at halfway houses have the ability to take care of themselves, maintain their personal hygiene, practice self-care with little effort to improve physical and mental health, and conduct simple economic activities [3]. A halfway house should be a place for patients with mental illnesses to receive rehabilitation before they return to their own family. Another study revealed that, compared with patients who did not receive community support rehabilitation (outpatient only), the acute hospitalization days and medical expenses required for patients with acute psychiatric illnesses were lower in patients living at halfway houses [4]. Another study indicated that residents of halfway houses exhibited better social function and economic benefits than did those in chronic psychiatric wards after community rehabilitation. Halfway houses have been included in health insurance coverage in Taiwan since 1995. In 2011, the principals of halfway houses were upgraded to the status of professionals. The number and hours of continuing education courses were included in the statutory provisions. The official budget consisted of start-up costs, equipment costs, and subsidized hospital bed charges [5].

The operational efficiency of halfway houses is important for the following reasons: (1) institutions can identify and compare assessments of good and poor implementation; (2) they can identify the most suitable scale to meet the service requirements of the group; and (3) they can integrate resources and effectively use human resources health policy formation [6]. Data envelopment analysis (DEA) has been commonly used in studies related to health care performance. For example, Gerard and Roderick [7] applied DEA to assess the England and Wales performance of hemodialysis satellite units (HDSUs) comparative efficiency in the delivery—DEA allows systematic and transparent investigation of efficiency improvements for individual HDSUs and setting of targets in relation to best practice peers. Shimshak and Lenard [8] observed that public institutions are less efficient than private ones through a study based on 40 nursing homes. Moreover, the quality variables have a significant impact on efficient execution; using DEA and Tobit regressions—a study determined that private institutions had better results than did public ones in a study of 22 nursing homes [9]. Applying DEA to 42 rural primary care centers, it was observed there were nine primary health care centers with technical efficiencies, five with scale efficiencies, and two with total efficiencies [10]. DEA was used to measure opportunity cost per Primary Health Trust in England including several health outcomes. It was found that the majority of health locations have the possibility of decreasing their expenditures between one percent and 15 percent without affecting outcomes [11]. However, a few studies have evaluated the performance of halfway houses. For example, in examining operational efficiencies using an input-oriented DEA for 50 halfway houses, eight nursing homes, and 32 shelters it was observed that private institutions were superior to public ones [12]. In the aforementioned studies, the decision-making units (DMUs) were evaluated together and divided into two categories, relatively efficient and relatively inefficient, by using the DEA model. In this assessment, the relative increase and decrease of inefficient DMUs does not affect the efficiency frontier or the proportion of relatively efficient DMUs. Relatively inefficient DMUs have no role in seeking direction for improvement.

The purpose of this study was to establish an advanced assessment mechanism for halfway houses by context-dependent DEA to measure their efficiency. The current study focuses on adjusting halfway houses to the most suitable mode under their existing economic scale. We apply context-dependent DEA to adjust the input and output based on the efficiency stratification and relatively inefficient DMUs. Input-oriented efficiency score is used because halfway houses generally want to obtain the maximum benefits under fixed existing resources. Thus, the relative attractiveness and progress are evaluated, and individual decision-making units’ competitive advantage and potential competitors are determined [13]. Context-dependent DEA can be used to discriminate the performance of efficient DMUs, and provide more accurate DEA results related to the performance of all DMUs [14]. Context-dependent DEA can measure the relative attractiveness and relative progress and thus evaluate DMUs with worse and better performance. When the DMUs in a particular level have equal performance, the attractiveness and progress enable their performance to be differentiated. The lowest efficiency of a particular assessment context (or third option) and the context-dependent DEA performance are determined not only according to the efficiency frontier but also based on the inefficient DMUs. Context-dependent DEA has made this approach more powerful and enabled the local and global identification of better choices [15]. We also used a case study in Taiwan to demonstrate the applicability of our proposed approach. The home-style and community-based housing experience is the foundation of community-based mental health care in Taiwan. To control quality and establish responsibilities at mental healthcare institutions, the Department of Health Executive Yuan in Taiwan established service quality indicators with the help of experts based on hardware and software facilities (including the use of technology facilities, for example safety equipment, alarm system, computer application, human resources management system etc.) of mental health institutions from 1985. Mental rehabilitation institutions have been formally appraised every three years since 2004. However, the current evaluation approach cannot provide the information about performance efficiency, which is important for funding decision of the government. Our proposed approach can bridge the gap and operators of halfway houses can apply it to provide residents with a good living environment and rehabilitation service quality.

## 2. Methodology

### 2.1. Variable Selection

The selection of DEA input and output variables is usually based on expert advice, experience, and economic theory. Input variables are those that reflect basic health requirements. The number of professionals in each institution, including psychiatrists, nurses, occupational therapists, social workers, and clinical psychologists, was considered one of the input variables. In similar studies of Chang and Cheng [16], Jehu Appiah et al. [17], Kundurjiev and Salchev [18], and Laine et al. [19], the performance accreditation of halfway houses, psychiatric hospitals, and nursing homes was listed as an indicator of input variables. In addition, the operating floor area of the organization, was also included in the input variables. These variables were also listed as variables in the performance of halfway houses and nursing homes [13,16]. The number of beds was the third input variable in this study. If the number of beds increases, the size of the institution’s hardware must also increase. These variables were listed as input indicators in the performance of psychiatric hospitals, nursing homes, and long-term care institutions [16,17,18], Balamatsis and Chondrocoukis [20].

To emphasize community health promotion, rehabilitation institutions must pay attention to patient safety and service quality, as well as help promote policies for the improvement of mental health services. The final result is a combination of cases and quality measurement. In the present study, human resources management, which completes the accreditation of mental rehabilitation institutions, is considered one of the output variables. Factors that were listed as performance indicators for halfway houses [12] in addition to management measures for the quality of rehabilitation services were also included in the output variables. These factors are similar to those adapted by Chang [21] and O’Neill [22] for the purpose of performance accreditation in general and for psychiatric hospitals and nursing homes. Problem solving ability was also included in the output variables. These factors are similar to those adapted by Osman [23] and Redfern et al. [24] for the purpose of performance indicators of nurses at hospital intensive care units. The measurements of the output variables, including management of human resource, management of rehabilitation service quality, and problem-solving ability, which were currently used by the Ministry of Health and Welfare in Taiwan are shown in Table 1.

The input variables used to assess the overall performance of rehabilitation institutions in this study were as follows: (1) the number of professional staff; (2) the business area (m2); and (3) the number of beds. The output variables were as follows: (1) management of human resource; (2) the management of rehabilitation service quality; and (3) problem solving ability. Therefore, a total of six variables were used as input and output variables in this study. This study included data from 38 halfway houses and mental health rehabilitation institutions that were evaluated by the Ministry of Health and Welfare of Taiwan in 2014.

### 2.2. Context-Dependent DEA

Context-dependent DEA was used to compare the attractiveness of DMUs. In context-dependent DEA, the evaluation contexts are obtained by partitioning a set of DMUs into several levels based on efficiency frontiers. Each efficiency frontier provides an evaluation context to measure the relative attractiveness. Although DMUs in a specific level are considered to have equal performance, the attractiveness measure enables this “equal performance” to be differentiated based on the specific evaluation context. A combination of attractiveness and progress measures can further characterize the performance of DMUs [25]. Suppose that N DMUs represent the examined halfway houses. Let *x_in_* and *y_jn_* be the amount of the *i_th_* input consumed and the amount of the *j_th_* output produced by the nth DMU, respectively. Consider the following linear programming (LP) problem:(1)$$\begin{array}{l} \phi *(l,k)=\underset{\lambda_{n},\phi (l,k)}{Min}\phi (l,k)\\ s.t.\\ \sum_{n\in F(C^{l})}\lambda_{n}\chi_{in}\leq \phi (l,k)x_{ik},\\ \sum_{n\in F(C^{l})}\lambda_{n}y_{jn}\geq y_{jk},\\ \phi (l,k),\lambda_{n}\geq 0;\;\forall i\;\mathrm{and}\;j,n\in F(C^{l}) \end{array}$$
where *F*(.) is the correspondence from a DMU set to the corresponding subscript index set, $C^{l}=\left\{DMU_{n},n=1,\ldots N\right\},C^{l+1}=C^{l}-D^{l},D^{l}=\langle DMU_{K}\in C^{l}|\phi (l,k)\rangle ,\;\mathrm{and}\;\phi *(l,k)$ is the optimal value of the LP problem when DMU*_k_* is under evaluation. Let D1 be the level 1 best practice frontier because it includes all of the frontier DMUs from the original input-oriented CCR model proposed by Charnes, Cooper and Rhodes in 1978. Based on the algorithm developed by Seiford and Zhu [13], we derived the lth-level best practice frontier by using Equation (1).

To rank the halfway houses in a specific level $D^{l\prime }$ based on their relative attractiveness scores, consider the following LP problem:(2)$$\begin{array}{l} G_{r}^{*}(a)=\underset{\lambda_{n},H_{r}(a)}{Min}G_{r}(a)\\ s.t.\\ \sum_{n\in F(D^{l\prime +a})}\lambda_{n}y_{jn}\leq G_{r}(a)x_{ir}\\ \sum_{n\in F(D^{l\prime +a})}\lambda_{n}y_{jn}\geq y_{jr}\\ G_{r}(a),\lambda_{n}\geq 0;\;\forall i\;\mathrm{and}\;j,n\in F(D^{l\prime +a}), \end{array}$$
where the *r*th halfway houses $DMU_{r}=(x_{ir},y_{jr})$ from a specific level $D^{l\prime }$ and the value of $G_{r}^{*}(a)$ is the attractiveness of *DMU_r_*. The output-specific attractiveness measure defines the difference between $DMU_{r}\in D^{l\prime }$ and $D^{l\prime +a}$ in terms of a specific output [26]. The *r*th halfway house is more attractive than the other investigated halfway houses if its $G_{r}^{a}(a)$ is higher than the others. The attractiveness measure can be used to (i) identify DMUs that have better performance and (ii) can rank DEA efficient DMUs [15]. Meanwhile, the progress measure of the *r*th halfway house *DMU_r_* can be obtained by the following linear programming problem [27]:(3)$$\begin{array}{l} H_{r}^{*}(b)=\underset{\lambda_{n},H_{r}(b)}{Min}H_{r}(b)\\ s.t.\\ \sum_{n\in F(D^{l\prime -b})}\lambda_{n}x_{in}\leq H_{r}(b)x_{ir}\\ \sum_{n\in F(D^{l\prime -b})}\lambda_{n}y_{jn}\geq y_{jr}\\ H_{r}(b),\lambda_{n}\geq 0;\;\forall i\;\mathrm{and}\;j,n\in F(D^{l\prime -b}), \end{array}$$

The progress measure of the *r*th halfway house is defined by $1/H_{r}^{*}(b)$. A smaller $1/H_{r}^{*}(b)$ value is preferred, because a higher $1/H_{r}^{*}(b)$ value indicates that more progress is expected for *DMU_r_*. The progress of a DMU is determined by considering DMUs with better performance in the evaluation context [15]. The progress scores mainly indicate the extent of improvement in productivity that is required to achieve a higher level of efficiency [28]. When the relative attractiveness of each class is higher than 1, the higher values indicate relative attractiveness with increased competitive advantage. On the other hand, when relative progressive values are greater than 1, the higher values indicate the relative progress from low relative efficiency; thus, the input–output allocation of resources must be adjusted to increase efficiency [29].

## 3. Case Study

### 3.1. Descriptive Analysis

A case study in Taiwan was selected to demonstrate the feasibility of the context-dependent DEA for the performance evaluation of halfway houses. Based on the data provided by the Psychological and Oral Health Division, Ministry of Health and Welfare of Taiwan, and Joint Commission of Taiwan, a total of 38 halfway houses were evaluated by the Ministry of Health and Welfare in Taiwan. The descriptive statistics for the 38 halfway houses in Taiwan are shown in Table 2. After the input and output variables were determined through the DEA analysis, the isotonicity condition was adjusted. Under these conditions, when input increases, output cannot be reduced [30]. In addition, the number of DMUs was twice the sum of the input and output items [31]. According to Coelli et al. [32], when the increase (decrease) of input causes the decrease (increase) of output, the efficiency accreditation of DEA may deviate. Therefore, before the first phase of accreditation, the current study assessed whether the linear relationship of two variables affected the isotonicity condition. In this study, the correlation coefficient between the inputs and outputs was measured using the Pearson correlation matrix. The following correlations were revealed: management of manpower and human resources: 0.257; management of manpower and rehabilitation service quality: 0.002; management of manpower and problem solving ability: 0.242; organization of the business area and human resources management: 0.232; management of area and rehabilitation services quality: 0.299; management of area and problem solving ability: 0.221; number of beds and human resource management: 0.245; number of beds and rehabilitation service quality: 0.138; and number of beds and problem solving ability: 0.285. No significant correlations, negative correlations, or deviations were observed. Therefore, these input and output variables indicated “isotonicity” relationships. The number of halfway houses must be triple the number of inputs and outputs. The 38 halfway houses used in this study were more than triple the number of inputs (3) and outputs (3) (i.e., 38 > 3(3 + 3) = 18). Therefore, the DEA model developed in the current study has high construct validity.

### 3.2. Performance Analysis

The overall technical efficiency scores for 38 halfway houses were calculated using an input-oriented DEA model. Overall technical efficiency (TE) of halfway houses was then analyzed by pure technical efficiency (PTE) and scale efficiency (SE). PTE or technical efficiency refers to producing the maximum amount of output from a given amount of input or alternatively, producing a given output with minimum input quantity. Scale efficiency (SE) is generally regarded as a separate efficiency issue, and refers to whether a unit is operating with optimal production size for producing a defined output and can be assessed in terms of production by referring to the notion of returns to scale (RTS). Increasing returns-to-scale (IRS) are said to exist when a proportional increase in inputs causes outputs to increase by a greater proportion, whereas decreasing returns-to-scale (DRS) is the situation in which an increase in inputs causes output to increase by a smaller proportion. SE can be evaluated by solving the DEA linear programming problem for TE under the assumption of constant returns to scale (CRS) and variable returns to scale (VRS). The measure of scale efficiency for unit *t* is as follows:SE*_t_* = Constant Return to Scale Technical Efficiency (CRSTE*_t_*) / Variable Return to Scale Technical Efficiency (VRSTE*_t_*).

The technical efficiency, scale efficiency, and the nature of returns to scale are presented in Table 3. The results indicate that the overall technical inefficiencies of halfway houses are primarily caused by pure technical inefficiencies instead of scale inefficiencies. From Table 3, the pure technical efficiency is 0.715, which shows that all halfway houses are maintained at the current output level, and the average input resource can be reduced by 28.5% to achieve the leading edge of efficiency. These findings also suggest that most managers should primarily focus on removing the technical inefficiencies in halfway houses, which will enable them to improve their scale efficiencies. In this study, five halfway houses achieved an SE of 1 (DMU 11, 12, 21, 24, 35), indicating that the current input resources have reached the most appropriate economic scale. Thirty halfway houses received modest returns (increasing returns), indicating that the halfway houses should expand to achieve their optimal size. Three halfway houses faced diminishing returns, indicating that the rate of increase for input was greater than that of output; thus, investment should be reduced. This model described five benchmarks (13.16% of the sample), and only 21 hallway houses achieved efficiency of ≥92.6%.

### 3.3. Constructing a Benchmark-Learning Roadmap

#### 3.3.1. Stratification

By incorporating the stratification DEA, attractiveness measure, and progress measure, a benchmark-learning roadmap was established to improve the inefficient halfway houses progressively and determine the most successful halfway house. Using the stratification DEA model from Equation (1) to (3), the three levels of efficient frontiers are presented in Table 4. The benchmark targets of the inefficient halfway houses at level 3 employ the halfway houses at level 2 (medium level) as an initial target to improve their efficiency in the first stage [25]. During the second stage, after the halfway houses at level 3 reach the level 2 efficiency frontier, they can use the level 1 efficiency frontier as a secondary benchmark for improvement. This composition of learning tracks for halfway houses at different levels is termed “benchmark-learning roadmap.” However, Chen et al. [14] observed that the levels obtained using Equation (3) do not necessarily follow the order of the TE scores. This is because we used the stratification DEA model, where the classification is based on the efficiency frontier and hence it is not easy to determine the split point with TE. For example, seven halfway houses (DMU 4, 5, 7, 18, 27, 32, 34) at level 2 may have larger TE scores than DMU 26 at level 1. A series of step-by-step benchmarks (or call benchmark-learning roadmap) allows inefficient halfway houses to learn and gradually improve their operational efficiency.

#### 3.3.2. Attractiveness and Progress

The results of the attractiveness and progress measures of the 38 halfway houses are presented in Table 5 with various efficiency frontiers as evaluation contexts. The number to the right of each score indicates the ranking determined by attractiveness and progress. Regarding the attractiveness measure, when level 2 was selected as the evaluation context, DMU 21 at level 1 was the most successful halfway house, with the highest attractiveness score of 2.216. The halfway houses at the first level can be ranked using the attractiveness measure in the order of DMU 21, 35, 24, 2, 22, 29, 31, 11, 3, 26, 1, and 8. When level 3 was selected as the evaluation context, DMU 21 remained the most successful halfway house, with the highest attractiveness score of 2.954, followed by DMU 35. These findings indicate that DMU 21 was the most attractive halfway house, regardless of the evaluation context.

Regarding the progress measure, when level 1 was selected as the evaluation context, DMU 14 was the worst halfway house at level 3, with the highest progress score of 0.617. The halfway houses at level 3 can be ranked using the progress measure. When level 2 was selected as the evaluation context, DMU 38 was the least successful halfway house at level 3. The ranking position changed for DMU 1, 3, 8, 22, 26, 29, and 31 at level 1, and DMU 12, 14, 36, 38, 9, 15, 17, and 37 at level 3, when the evaluation context was changed. This demonstrated that the performance of halfway houses can be dependent on their evaluation background [33].

The performance of halfway houses that are not located in the levels 1 or 3 can be characterized through their attractiveness and progress scores [34]. To understand its own merits and demerits, DMUs can highlight their competitive advantages in the industry by merging the relative attractiveness and relative progress values to detect whether the current business strategy and resource allocation is appropriate. To indicate the competitive advantages and disadvantages more clearly, four quadrants were delineated based on the average of the attractiveness and progress value of 15 DMUs at level 2, because the second level can distinguish its competitive advantage position in the industry by merging the relatively attractive value and the relative progress value to detect whether the existing management strategy and resource allocation are appropriate.

In Figure 1, each halfway house at level 2 was classified into a quadrant by examining whether the attractiveness score was higher or lower than 1.365 and whether the progress score was higher or lower than 0.676. The attractiveness and progress scores are presented in a two-by-two matrix to classify the halfway houses at level 2. The first quadrant had low relative attractiveness value, but high relative progress value, indicating that a close competitor existed in the current competitive position and that the halfway house must significantly improve its management strategies for resource allocation to reach the ideal efficiency frontier. The second quadrant had high relative attractiveness value and high relative progress value. Even though there are no close competitors, the halfway house must work harder to improve its performance by increasing the attractiveness value and reducing the progress value. The third quadrant had low relative attractiveness value and low relative progress value, indicating that although a potential competitor exists, it will be relatively easy to catch up with the DMUs in the higher level. However, managers must also check whether the current strategy is appropriate by evaluating their attractiveness value. The fourth quadrant had high relative attractiveness value and low relative progress value, indicating that these DMUs are in a better position for competition, have relatively strong performance, and can maintain the current management principles. Relatively good performance for a DMU indicates high relative attractiveness value and low relative progress value. Therefore, DMUs located in the fourth quadrant had the strongest performance, whereas those in the first quadrant had the weakest performance. DMUs in the second and third quadrants had both high and low values. DMUs in the second quadrant can reduce their progress value, whereas those in the third quadrant should enhance their attractiveness value to reach the more competitive fourth quadrant. In short, 38 halfway houses were split into four zones in this study:

Zone LH: The halfway house included in this zone was DMU 19. The findings indicate that the halfway house located in Zone LH has better competitive advantages than those at level 2.

Zone HH: This zone included the halfway houses DMU 4, 5, 7, 20, 27, and 34. These DMUs should place more emphasis on activities to improve their output substantially. For example, these halfway houses should increase the community resource service and activity to attract more customers.

Zone LL: The five halfway houses included in this zone were DMU 6, 23, 25, 28, 30, and 33. These halfway houses should establish short- or medium-term plans to enhance their competitive advantage to move into Zone LH. For example, these halfway houses should adopt facilities to enhance their competitive advantage.

Zone HL: This zone included the DMU 18 and 32 halfway houses. These halfway houses should increase their efforts toward learning more capabilities for effective outcomes, such as enhancing the activities of operational management and relocating the resources between input and output. Furthermore, these halfway houses must build short- or medium-term plans to enhance their competitive advantages.

## 4. Discussion 

The clinical indicators of patient-levels and the outcomes of long-term care (such as quality of life) are the relevant output variables for this study, but they are not readily available. Through the assistance of the Ministry of Health and Welfare of Taiwan, we successfully obtained the appraised results of the halfway houses as the output variables data of our case study. The mental healthcare system includes very few performance indicators that can assess efficiency, costs, or expenditures. It is also particularly difficult to obtain relevant information from healthcare costs and expenditures, and data regarding healthcare expenditures for patients with diseases are often limited by scope and comparability. Therefore, a combination of cases, quality measurements, representative factors, and experience are required to provide an accurate picture of the efficiency of mental healthcare [35]. The present study provided a verification model of the operating performance of halfway houses. The DMU 21 halfway house had the best benchmark model (CRSTE, CRSPTE, and SE of 1, level 1, attractiveness rank of 1) and was the most suited to patient needs and economic conditions (results are presented in Table 2, Table 3 and Table 4). DMU 21 also obtained the highest score among psychiatric rehabilitation institutions from the Ministry of Health and Welfare in 2014. 

Through DEA analysis, the study revealed that five halfway houses achieved the most economical scale. Thirty halfway houses that belonged to the increasing payoff type should be oriented toward increasing output. For example, these halfway houses can strengthen the organization responsible for operations and management training, provide a full-time staff salary stability system, arrange an appropriate day and night manpower configuration, plan and regularly modify the activities in the management of rehabilitation services, increase the rights and interests of residents, enhance the satisfaction of residents and their families, and improve health maintenance measures. Three halfway houses had diminishing returns and thus require part-time employees or project co-operation models to reduce manpower; they can also use multipurpose areas to reduce free space, or use external space. These halfway houses must also reduce empty beds or increase bed occupancy. The three halfway houses with diminishing returns had higher investments but low returns. Therefore, to avoid closure, these institutions must consider basic efficiencies to improve returns.

## 5. Conclusions

In this study, context-dependent DEA was used to compose a successful benchmark-learning roadmap for halfway houses and improve the inefficient DMUs progressively. This study also identified the most successful DMU. DMUs at level 2 have the benefit of learning from level 1 and being attracted to level 3. This will enable DMUs to analyze their location in the market and implement methods to improve efficiency and competitiveness. Compared with a traditional two- stage positive mode, which has been employed for large-scale intelligence screening of military personnel, two-stage window screening in this study (combined two-stage positive and two-stage negative) indicated increased accuracy and economic benefits in mental healthcare institutions [36]. Therefore, context-dependent DEA can be used to efficiently allocate the resources of halfway houses and community resources to improve the growth of halfway houses. The limited resources can be used to further develop incentives for halfway houses with attractive and progressive value. The results of this study can help halfway houses to achieve their purpose, which is to reduce the institutionalization and waste of medical resources caused by the long-term hospitalization of patients with mental illnesses.

The data used in this study were obtained from the Taiwan Ministry of Health and Welfare. Due to the lawful protection of personal medical data, the study could not access data regarding the names and regions of the halfway houses. Therefore, the characteristics specific to region and institutional ownership (public or private) could not be analyzed. Data regarding the incomes of halfway houses was derived from the National Health Insurance Administration Taiwan Ministry of Health and Welfare. The halfway houses received fixed funding based on the number of services (beds) during the assessment; therefore, this factor was not included in the study. Future studies can explore comparisons of halfway houses based on characteristics specific to region, institutional ownership (public or private), and business improvement measures.

In conclusion, the study findings provide several references for improvements in halfway houses, and determine the relative attractiveness and progress value of each halfway house in order to understand the learning object, and the adopted operation strategy. These findings can help operators of halfway houses to ensure cost-effectiveness and to configure methods to improve the institutions. The findings can also help the Medical Joint Commission understand how to modify the benchmark and regulations for accreditation while policy makers will be able to grasp how to appropriately allocate resources to maximize effectiveness.

## Figures and Tables

**Figure 1 healthcare-08-00189-f001:**
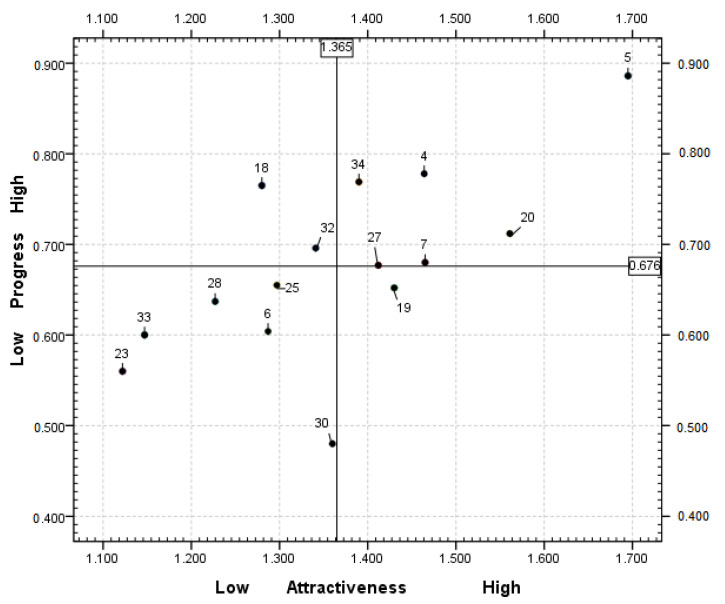
Attractiveness and progress scores for the halfway houses at the second level.

**Table 1 healthcare-08-00189-t001:** Measurements of output variables.

Output Variable	Measurement
Human resource management	The management of the person in charge of the organization. The stability of the manpower of the full-time and part-time staff The supervision system The appropriate allocation of day and night manpower
Rehabilitation quality management	Appropriate project closure Work manual Complete and proper management of records Proper planning and regular revision of activities Appropriate rehabilitation fund management Protection measures of residents’ rights Residents’ property storage Residents’ freedom of entry and exit Satisfaction of residents and their families Health maintenance measures Convening quality management related meetings
Problem solving ability	Establish procedures for handling emergency Establish procedures for handling medical and abnormal incidents Implement and establish the organization’s emergency response management mechanism

**Table 2 healthcare-08-00189-t002:** Descriptive statistics of the case study.

**Inputs**	**Mean**	**Minimum**	**Maximum**	**Std. Dev.**
Square meters of area	410.93	216.00	927.60	156.17
The volume of services	32.47	18.00	90.00	11.99
Professionals	5.00	2.50	13.00	1.73
**Outputs**	**Mean**	**Minimum**	**Maximum**	**Std. Dev.**
Human resource management	7.96	5.00	10.80	1.56
Rehabilitation quality management	20.30	14.20	25.00	2.46
Problem solving ability	2.71	0.80	4.60	1.10

**Table 3 healthcare-08-00189-t003:** Efficiency scores based on input-oriented scores.

DMU	CRSTE	VRSPTE	SE	RTS	DMU	CRSTE	VRSPTE	SE	RTS
1	0.687	0.821	0.836	irs	20	0.712	0.718	0.992	irs
2	0.94	0.983	0.957	irs	21	1.000	1.000	1.000	-
3	0.868	0.907	0.957	irs	22	0.780	0.782	0.997	irs
4	0.778	0.784	0.993	irs	23	0.560	0.668	0.838	irs
5	0.886	0.959	0.924	drs	24	1.000	1.000	1.000	-
6	0.604	0.940	0.643	irs	25	0.655	0.743	0.882	irs
7	0.680	0.715	0.951	irs	26	0.658	0.677	0.971	irs
8	0.704	0.742	0.948	irs	27	0.677	0.722	0.938	irs
9	0.388	0.442	0.878	irs	28	0.576	0.621	0.928	irs
10	0.288	0.294	0.978	irs	29	0.759	0.776	0.978	drs
11	0.667	0.667	1.000	-	30	0.647	0.659	0.982	irs
12	0.500	0.500	1.000	-	31	0.716	0.749	0.956	irs
13	0.464	0.475	0.976	irs	32	0.696	0.728	0.956	irs
14	0.617	0.656	0.941	irs	33	0.550	0.621	0.886	irs
15	0.519	0.528	0.984	irs	34	0.768	0.896	0.857	irs
16	0.411	0.653	0.629	irs	35	1.000	1.000	1.000	-
17	0.467	0.532	0.877	irs	36	0.480	0.486	0.987	irs
18	0.765	0.778	0.984	irs	37	0.478	0.510	0.937	irs
19	0.653	0.760	0.859	drs	38	0.549	0.686	0.801	irs
					Mean	0.662	0.715	0.926	

**Table 4 healthcare-08-00189-t004:** Levels of decision-making units (DMUs).

First Level		Second Level		Third Level	
DMU No.	TE	DMU No.	TE	DMU No.	TE
1	0.687	4	0.778	9	0.388
2	0.940	5	0.886	10	0.288
3	0.868	6	0.604	12	0.500
8	0.704	7	0.680	13	0.464
11	0.667	18	0.765	14	0.617
21	1.000	19	0.653	15	0.519
22	0.780	20	0.712	16	0.411
24	1.000	23	0.560	17	0.467
26	0.658	25	0.655	36	0.480
29	0.759	27	0.677	37	0.478
31	0.716	28	0.567	38	0.549
35	1.000	30	0.647		
		32	0.696		
		33	0.550		
		34	0.768		

**Table 5 healthcare-08-00189-t005:** Attractive and progress scores of the halfway houses.

Evaluation Context			Evaluation Context			Evaluation Context		
First Level	Second Level	Third Level	Second Level	First Level	Third Level	Third Level	First Level	Second Level
DMU	1st-Degree ^a^	2nd-Degree ^a^	DMU	1st-Degree ^b^	1st-Degree ^a^	DMU	1st-Degree ^b^	2nd-Degree ^b^
21	2.216(1) ^c^	2.954(1)	4	0.778(14)	1.464(4)	12	0.500(8)	0.836(9)
24	1.643(3)	2.194(3)	5	0.886(15)	1.695(1)	14	0.617(11)	0.844(10)
35	1.682(2)	2.235(2)	7	0.680(10)	1.465(3)	36	0.480(7)	0.802(8)
1	1.082(11)	1.443(10)	18	0.765(8)	1.280(12)	38	0.550(10)	0.888(11)
2	1.239(4)	1.717(4)	19	0.652(6)	1.430(5)	9	0.389(2)	0.729(6)
3	1.187(9)	1.543(7)	20	0.712(12)	1.561(2)	13	0.464(4)	0.673(4)
8	1.081(12)	1.437(11)	27	0.677(9)	1.412(6)	15	0.520(9)	0.479(2)
11	1.130(8)	1.506(8)	32	0.696(11)	1.341(9)	16	0.411(3)	0.640(3)
22	1.199(5)	1.575(6)	34	0.769(13)	1.390(7)	17	0.467(5)	0.792(7)
26	1.111(10)	1.456(9)	6	0.604(4)	1.287(11)	37	0.478(6)	0.684(5)
29	1.192(6)	1.619(5)	23	0.560(2)	1.122(15)	10	0.288(1)	0.425(1)
31	1.051(7)	1.426(12)	25	0.655(7)	1.297(10)			
			28	0.637(5)	1.227(13)			
			30	0.480(1)	1.360(8)			
			33	0.600(3)	1.147(14)			

Note: ^a^ This represents attractiveness; ^b^ This represents progress; ^c^ Ranks are given in parenthesis.

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
