# Peer review of "Improving Efficiency Assessment of Psychiatric Halfway Houses: A Context-Dependent Data Envelopment Analysis Approach"

_healthcare, 2020, doi:10.3390/healthcare8030189_

Round 1

Reviewer 1 Report

I appreciate the effort of Shen et al. to answer an important question, in pursuit of improving care and outcomes for a vulnerable population. I offer the following major and minor concerns in hopes that the presentation of their study may be improved so that readers will be able to appreciate and learn from their endeavors.

MAJOR:

  • The organization of and language throughout the Introduction could be improved. As a reader, I found that I had to work to understand what was being communicated and how it relates to the purpose or motivation of the study. The Introduction of both halfway houses and DEA could be more focused. I worry that without revision that readers will not persist to appreciate what the study has accomplished.
  • I don’t understand how the output variables were measured: “(1) management of human resource; (2) the management of rehabilitation service quality; and (3) problem solving ability.”
  • I’m also left wondering if the selected output variables are the right ones. Do the selected output variables capture or reflect meaningful outcomes for patients? If not, why are they chosen? If so, how so?
    • In the discussion, the authors state, “The clinical indicators of patient-levels and the outcomes of long-term care (such as quality of life) are the relevant output variables for this study, but they are not readily available” (lines 319-320). This gap is a major problem for this study and must be better addressed. There is a potential danger in having delivery systems target outcomes that are not the relevant ones. The authors must address how setting the variables that they were able to incorporate as target outcomes will lead to desired improvements. Is there any literature linking their selected output variables with the relevant patient-centered outcomes?
  • Though the authors conclude, "the study findings can provide several references for improvements in halfway houses" (lines 367-368), it is unclear what actions the halfway houses or system as a whole would need to take to improve. Rather the authors make relatively vague statements, such as:
    • “Even though there are no close competitors, the halfway house must work harder to improve its performance” (lines 287-288).
    • “However, managers must also check whether the current strategy is appropriate” (lines 290-291).

MINOR:

  • Overall the organization of the paper which often weaves prior literature, presentation of this study, and interpretation of results together makes the paper difficult to read and appreciate. Here are two examples of how the presentation of results often weaves interpretation into the results:
    • “The results 230 indicate that the overall technical inefficiencies of halfway houses are primarily caused by pure 231 technical inefficiencies instead of scale inefficiencies. These findings also suggest that most managers 232 should primarily focus on removing the technical inefficiencies in halfway houses, which will enable 233 them to improve their scale efficiencies” (lines 230-233).
    • “According to Morita, Hirokawa, and Zhu [25], the benchmark targets of the inefficient halfway 249 houses at level 3 employ the halfway houses at level 2 (medium level) as an initial target to improve 250 their efficiency in the first stage. During the second stage, after the halfway houses at level 3 reach 251 the level 2 efficiency frontier, they can use the level 1 efficiency frontier as a secondary benchmark 252 for improvement. This composition of learning tracks for halfway houses at different levels is 253 termed “benchmark-learning roadmap.” However, Chen et al. observed that[14] the levels obtained 254 using Equation (3) do not necessarily follow the order of the TE scores. For example, seven halfway 255 houses (DMU 4, 5, 7, 18, 27, 32, 34) at level 2 may have larger TE scores than DMU 26 at level 1” (lines 248-255).

Reviewer 2 Report

Thank you for giving me the opportunity to review this paper.

The paper is easy to understand and the contribution is decent.

I have the following suggestions:

  1. The text is rather compact; please split your text in several parts.
  2. Why four zones? and why three levels in Table 3?
  3. You should explain why choosing an input-oriented efficiency score? why not a directional distance function?
  4. Make clear in your tables that we read input-oriented-efficiency.

Round 2

Reviewer 2 Report

Proofreading is needed.